# Analysis of the External and Internal Load in Wheelchair Basketball Considering the Game Situation

Víctor Hernández-Beltrán [1,*], Sergio J. Ibáñez [1], Mário C. Espada [2,3,4] and José M. Gamonales [1,5,6,*]

1. Training Optimization and Sports Performance Research Group (GOERD), Faculty of Sport Science, University of Extremadura, 10005 Cáceres, Spain; sibanez@unex.es
2. Instituto Politécnico de Setúbal, Escola Superior de Educação, 2914-504 Setúbal, Portugal; mario.espada@ese.ips.pt
3. Life Quality Research Centre (CIEQV-Leiria), 2040-413 Rio Maior, Portugal
4. Interdisciplinary Centre for the Study of Human Performance (CIPER), Faculdade de Motricidade Humana, Universidade de Lisboa, 1499-002 Cruz Quebrada, Portugal
5. Facultad Ciencias de la Salud, Universidad Francisco de Vitoria, 28223 Madrid, Spain
6. Programa de Doctorado en Educación y Tecnología, Universidad a Distancia de Madrid, 28400 Madrid, Spain
* Correspondence: vhernandpw@alumnos.unex.es (V.H.-B.); martingamonales@unex.es (J.M.G.)

**Featured Application: Quantifying the external and internal load of the wheelchair basketball will allow the coaching staff to analyze the physical demands of the players during training.**

**Abstract:** The systematic and programmed control and evaluation of the external and internal load of high-performance athletes during training sessions and high-level competitions allows the coaching staff to know a great amount of information to evaluate the physical condition of the players and the tactical positioning as well as to identify optimal performance. Therefore, this study aimed to analyse the external and internal load of wheelchair basketball players during training and competition matches considering the functional classification of the players. WIMU PRO[TM] inertial devices were used to collect the data. As independent variables, the sporting context and the functional classification were selected, and as dependent variables, the variables related to the external and internal load were established. The results reported significant differences between the contexts, with higher values in the competition than in the training sessions considering the internal and external load. Taking into account the functional classification, it is shown that the player with a higher functional classification obtained more differences and higher values. Understanding the external and internal load of the players during training and competitions is very important to personalise the training load according to the demands of the competition. It allows development of a progressive and modular training programme of loads to obtain the best performance. Consequently, injury risk of players due to overload will also be reduced.

**Keywords:** parasport; functional classification; matches; training sessions; inertial devices

## 1. Introduction

Inertial devices (ID) have been widely used in professional sports to quantify external load (EL) and internal load (IL) demands [1], because they allow characterization and quantifying of the players' performance levels [2]. The ID allows the measurement of different aspects of players' physical demands such as speed or distance covered in sprinting [3], as well as identifying the acceleration (Acc) and deceleration (Dcc) profiles of U18 female players [4] or speed and height in vertical jumps [5]. Likewise, these devices have been used in paralympic sports, such as football for blind people in order to quantify the distance, Acc, and Dcc during competition [6]. This technology allows coaches and players to develop their work in the best possible way, due to the large amount of real-time information linked to the physical condition of the players, allowing coaches to analyse

and evaluate the evolution of physical condition during training and competition [7]. In this line, it is recommended to quantify and evaluate the physical fitness of the player from the pre-season in order to prevent and reduce injury risk related to overload [8].

The use of IDs has allowed a specific quantification of the load, identifying possible imbalances or differences in the absorption of loads or impacts in conventional basketball players [9] through the analysis of asymmetries [10]. In the same way, these IDs have been used in different sports modalities to quantify the physical demands in handball and beach handball [11] or identify the influence of the use of constraints on the physiological responses of athletes through the manipulation of the field size or the presence of the goal-keeper in small-sided games [12]. Understanding the workload of practitioners in training and competition is very important in aiming for performance enhancement; however, there are different methodologies for quantifying the player's load or fitness condition, such as the use of physical tests [13] and instruments for quantifying the subjective load [14]. Therefore, it is necessary to select the proper instruments for quantifying the load considering the specificities of each sport and game position in order to obtain valid and reliable results [15,16].

In Wheelchair Basketball (WB), few studies used IDs for the analysis of players' physical demands [17]. In contrast, some studies have used heart rate (HR) or the rating of perceived exertion (RPE) scale [18] to quantify the IL of the players. Both methods are reliable and valid for monitoring the IL of the players [19]. HR analysis can be beneficial and supplementary for the design and creation of personalised exercises focused on the demands and needs of WB players, with high-intensity sessions being the most beneficial [20]. Bearing in mind the IL of the players, some studies analysed the HR during training sessions [21] or matches [22]. The subjective EL can be measured through the use of observational instruments and evaluate the load of the task regarding the game situation, the type of task, or the game space [23]. According to these studies, the WB players support great physical and physiological load during the season [20]. The use of these instruments for quantifying and evaluating the EL and IL of the players allows the coaches to identify the exercise thresholds as well as prevent player injuries [24]. Information regarding the EL and IL influence on the players' sport performance and how it progresses throughout the season is an important factor for the coaching staff in order to adjust the training loads and achieve the improvements desired [25].

WB is a sport discipline designed for people with disabilities (e.g., spinal cord injury, amputees, spina bifida, joint and muscle limitations, etc.), which affect their ability to run, pivot, or jump [26]. The athletes with these types of injuries have a limitation in motor control under the spinal cord injury, affecting the circulation system and loss of sweating capacity [27]. In the case of amputee athletes, they have a bigger capacity for movement due to their disability, which does not influence directly their performance because they only have a total or partial lack of the lower limbs, presenting a bigger functional level [28]. According to the trunk movement, control, and balance, a functional classification (FC) will be attributed to each player [29] with different scores, from 1.0, those players with lower functional capacity, to 4.5, those players with higher functional movement capacity [30]. This will directly influence the performance of the players [29,31]. Therefore, it is necessary to identify the loads that a WB player supports during training sessions and competition, as well as to consider the influence of the FC on the performance. Hence, this study aimed (a) to identify the physical demands of WB players in $5 \times 5$ situations considering the FC and (b) to identify differences in the EL and IL values of WB players according to the FC during training and competitive matches.

## 2. Materials and Methods

### 2.1. Design

The present study was performed under a cross-sectional design [32] because data collection was conducted at a specific time during the pre-season. An associative and

descriptive strategy was pursued [33], with the aim of analysing the EL and IL of WB players during training and competition.

### 2.2. Participants

A national WB team was involved in the study (*n* = 12; age = 31.5 ± 8.3 years, height = 167.0 ± 0.25 cm, weight = 68.0 ± 12.6 kg). In addition, considering it is a sport for people with disabilities, they have a CF according to the range of movements and trunk stability from FC 1.0 to FC 4.5. Table 1 shows demographic information about the subjects who made up the sample.

**Table 1.** Descriptive analysis of the participants.

| Players | Age (Years) | Weight (kg) | Height (cm) | Experience (Years) | FC |
|---|---|---|---|---|---|
| 2 | 41 ± 0 | 82 ± 2.82 | 181.5 ± 4.94 | 6.5 ± 2.12 | 1.0 |
| 1 | 39 | 68 | 182 | 5 | 2.0 |
| 4 | 31 ± 1.16 | 61 ± 6.97 | 146.5 ± 33.39 | 5.2 ± 2.87 | 2.5 |
| 2 | 25 ± 2.82 | 57.5 ± 21.92 | 167.5 ± 20.50 | 4 ± 1.41 | 3.0 |
| 3 | 28.3 ± 7.57 | 75 ± 8.88 | 180 ± 10.81 | 3.6 ± 1.54 | 4.0 |

kg: kilograms; cm: centimeters; FC: functional classification.

Eligibility Criteria

To include the players in the study, they must fit some inclusion criteria:

- Officially belonging to the national WB team and being part of a professional WB team.
- Having participated in at least 70% of the training sessions analysed
- Having played at least fifteen minutes in both matches.

Only the players who fit these criteria at the end of the data collection were included in the data analyzed.

### 2.3. Sample

The sample analysis consisted of all the tasks performed in training sessions and two official matches played during the period of analysis. To carry out the analysis between training and competition, only 5 × 5 tasks were selected. These tasks developed simulated games to obtain physical demands similar to the official matches, to compare the simulated full game in training with official matches, and to quantify the EL and IL of the players in each of the situations assuming the FC. Finally, 350 cases were selected for the analysis.

### 2.4. Procedures

Firstly, managers and directors of the national team were contacted to establish a first contact and explain the benefits and disadvantages derived from the study. After their approval, a meeting was held with the entire group of players to inform them about the procedures and the materials to be used during the data collection. Afterwards, informed consent was signed by all the members of the team. After a familiarisation session with the ID and the HR bands, all the tasks were monitored to quantify the EL and IL of the players during training sessions and two official matches. The study was developed under the premises of the Declaration of Helsinki [34], being approved by the Bioethics Committee of the University of Extremadura (Registration number 79/2022) and developed under the ethical standard for sports science research [35].

### 2.5. Instruments

Twelve WIMU PRO[TM] devices (RealTrack Systems, Almería, Spain) were used to evaluate the EL of the players. The IDs were placed in the interscapular area using an anatomical harness. HR bands (Garmin[TM], Olathe, Kansas) were used to quantify the IL of the players.

For collecting the data, the Ultra-Wide Band (UWB) system was used using 8 antennas placed around the basketball court [8]. To visualise the data in real-time during the sessions,

SVIVO^TM software (RealTrack Systems SL, v.2020, Almeria, Spain) was used through Advanced and Adaptive Network Technology (ANT)+ technology. Finally, SPRO^TM software (RealTrack Systems SL, v. 990, Almeria, Spain) was used for data processing and analysis.

### 2.6. Variables

The independent variables were:
- The context where the data collection was performed (training sessions or matches).
- The FC of each player.

The dependent variables were distance, explosive distance, Acc, Dcc, high-speed running (HSR), maximum Acc, maximum Dcc, maximum speed, average speed, player load (PL, impacts, average HR, maximum HR, and % maximum HR). These variables were normalised to the same time unit (minute) to properly conduct a comparison considering the independent variables. Previously, these variables were used in different studies in order to quantify the EL and IL of the players in different sports modalities [8,36,37].

### 2.7. Statistical Analysis

The Kolgomorov–Smirmov test [38] was used to analyse the sample distribution, obtaining a value higher than $p < 0.05$. Therefore, parametric tests were used for data analysis and hypothesis testing. Subsequently, a descriptive analysis of the sample was carried out to characterise the data (mean and standard deviation).

Student's $t$-test was carried out to identify the differences between the contexts analysed. Secondly, the sample was divided assuming the FC of each player and again analysed regarding the context. Similarly, the sample was analysed considering the type of task and game situation, and FC was added as a co-variable throughout the analysis. For the analysis and interpretation of the Effect Size (ES) of the comparative analysis, Cohen's d was used, following Hopkins et al.'s proposal [39]: low effect (0–0.2), small effect (0.2–0.6), moderate effect (0.6–1.2), high effect (1.2–2.0), and very high effect (>2.0). The ES and sample power were determined using GPower (v.3.1.9, University of Kiel, Kiel, Germany), from the data extracted from the Student's $t$-test and specifying a security level of $\alpha = 0.05$ [40]. Finally, the Z-Score test was used in order to compare the punctuation of the different variables according to each FC [38].

The Statistical Package for the Social Sciences software (version 27, 2021; IBM Corp., IBM SPSS Statistics for MAC OS, Armonk, NY, USA) was used to perform the different statistical analyses. The level of significance was set at $p < 0.05$.

## 3. Results

Table 2 shows the descriptive results according to the context regarding the dependent variables selected. The results reported which variables present differences depending on the variables studied and the ES of the comparison.

Secondly, through the use of normalized values, the differences between the context and the FC are revealed. The results reported that the players with FC 1.0 present differences in 8 out of 14 analysed variables, considering the EL and IL of the players (Figure 1a). Regarding the players with FC 2.0, the results highlighted only significant differences in three variables (Acc/min, Dcc/min, and average HR (Figure 1b)). Players with FC 2.5 had differences in most of the variables, except HSR/min and % maximum HR and HSR/min (Figure 1c). Considering the players with FC 3.0, the study revealed variations in all the objective IL variables, and, observing the rest of the dependent variables, only 6 out of 10 reported significant differences (Figure 1d). Regarding the players with a higher FC (4.0), the results revealed differences in the kinematics and neuromuscular EL variables and the % maximum HR (Figure 1e) (please see Supplementary Materials Table S1–S5).

**Table 2.** Descriptive analysis and differences between the contexts.

| Variables | | Training | | Match | | t | df | p | ES |
|---|---|---|---|---|---|---|---|---|---|
| | | $\bar{X}$ | SD | $\bar{X}$ | SD | | | | |
| Kinematics External Load | Distance/min (m) | 63.429 | 14.017 | 95.484 | 43.581 | −6.861 | 190 | 0.000 * | 0.990 |
| | Explosive distance/min (m) | 5.375 | 1.985 | 8.143 | 4.823 | −5.200 | 190 | 0.000 * | 0.751 |
| | Acc/min (m/s$^2$) | 26.264 | 4.754 | 75.351 | 32.837 | −14.496 | 190 | 0.000 * | 2.092 |
| | Dcc/min (m/s$^2$) | 22.387 | 5.876 | 75.799 | 33.244 | −15.502 | 190 | 0.000 * | 2.237 |
| | HSR/min (m) | 5.755 | 4.937 | 1.262 | 4.976 | 6.280 | 190 | 0.000 * | 0.906 |
| | Maximum Acc (m/s$^2$) | 3.663 | 0.962 | 3.165 | 0.638 | 4.221 | 190 | 0.000 * | 0.609 |
| | Maximum Dcc (m/s$^2$) | −3.817 | 0.964 | −3.504 | 0.655 | −2.630 | 190 | 0.009 * | 0.380 |
| | Average speed (km/h) | 5.238 | 0.470 | 4.019 | 1.426 | 7.951 | 190 | 0.000 * | 1.148 |
| | Maximum Speed (km/h) | 15.242 | 2.159 | 12.481 | 4.247 | 5.677 | 190 | 0.000 * | 0.819 |
| Neuromuscular External Load | PL/min (a.u.) | 0.734 | 0.214 | 1.481 | 1.115 | −6.445 | 190 | 0.000 * | 0.930 |
| | Impacts/min (n) | 156.050 | 70.753 | 234.150 | 178.912 | −3.977 | 190 | 0.000 * | 0.574 |
| Objetive Internal Load | Average HR (bpm) | 118.916 | 23.689 | 133.833 | 21.830 | −4.526 | 190 | 0.000 * | 0.655 |
| | Maximum HR (bpm) | 152.295 | 30.525 | 160.750 | 24.798 | −2.102 | 190 | 0.037 * | 0.304 |
| | % maximum HR (bpm) | 68.438 | 12.838 | 69.492 | 11.832 | −0.590 | 190 | 0.556 | 0.085 |

$\bar{X}$: Mean, SD: standard deviation; * $p < 0.05$; ES: effect size; Acc: acceleration; Dcc: deceleration; HSR: high-speed running; HR: heart rate; PL: player load.

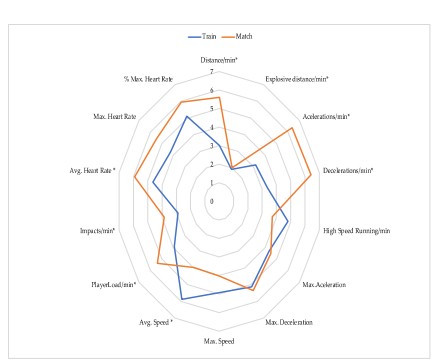

(**a**) Comparison of the FC 1.0 players

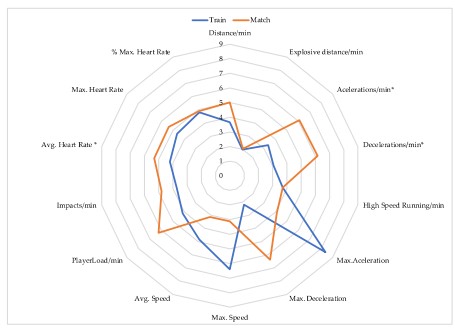

(**b**) Comparison of the FC 2.0 players

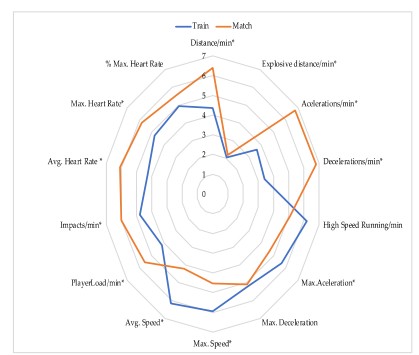

(**c**) Comparison of the FC 2.5 players

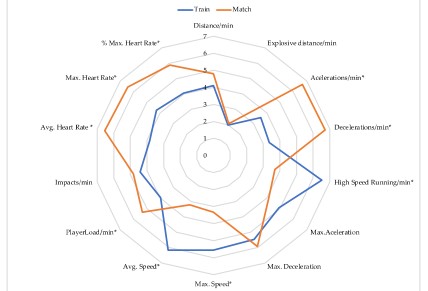

(**d**) Comparison of the FC 3.0 players

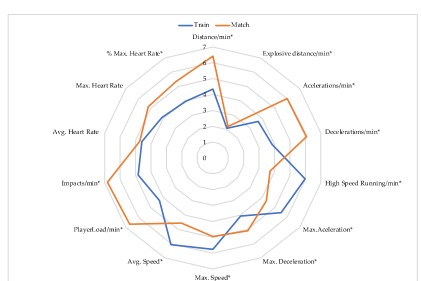

(**e**) Comparison of the FC 4.0 players

**Figure 1.** Comparison of the EL an IL of the players regarding the context analysed. * $p < 0.05$.

## 4. Discussion

The main purpose of this study was to analyse the EL and IL of WB players in different contexts (training and matches), as well as to analyse whether the FC influences the sport performance and if there are differences between the FC. The results reported significant differences between the contexts according to the analysed variables. The number of Dcc/min was the variable with the highest ES, which means that there are higher differences between the context than the other variables. On the other hand, the % of maximum HR was the only variable that did not present significant differences between the

contexts, being higher in competitions. Likewise, the IL variables were higher in matches than in training sessions.

According to the neuromuscular EL variables, it was observed that the values were higher in matches compared to training sessions. Analysing training sessions and matches is very important for the coaching staff in order to understand the physical demands during those contexts and adapt the training sessions to the competitive demands, which is the most demanding scenario in sports and parasports [41]. Considering the FC separately, the player with higher punctuation (FC 4.0) obtained higher values in the majority of the variables, a fact that we attribute to the fact that it was the players with the largest trunk range of movements and functional capacity. These results are related to the sitting height because the player with a higher height obtains better performance in field tests [26]. Thus, the coaching staff must develop tasks to improve speed, mobility, agility, and strength, as well as tactical and technical skills to improve players' performance in matches.

Considering the analysed context (training or competition), according to the kinematic variables, it was observed how four out of nine variables presented higher values in competition compared to training. For example, the average and maximum speed was higher in training, because the defensive process is less aggressive than in competition, which allows fast transition and attacks. These results are related to the study carried out by Vučković et al. [42] in conventional basketball, in which the guards and centres were associated with the lowest speed zones. On the other hand, distance/min, explosive distance/min, Acc/min, and Dcc/min, were the variables with higher values in competitions than in training, with these results highlighting the intermittent nature of the sport [20]. Therefore, the number of Acc and Dcc is a factor that determines the performance of the player [43]. Similarly, the distance travelled is high due to the high displacement capacity of the players, as well as the ability to turn and change direction, which are considered Acc and Dcc.

In relation to the IL variables, it was observed that the values are slightly higher in matches. The differences are small; however, these results are similar to those found by Pérez et al. [44] in which the range of HR was between 131 and 152 beats per minute. Similarity, Conners et al. [45] reported that the average HR was around 140 beats per minute. Regarding the % maximum HR, this value was around 69%, so it is demonstrated that the game is very demanding considering the cardiovascular requirements [46]. Thus, the results obtained in this study are supported by previous studies in WB, despite that research in this parasport is scarce compared to conventional basketball and even other parasports. To improve the player's cardiovascular condition, we suggest that it is important to carry out training tasks aiming to improve the aerobic condition and consequently the global fitness condition and recovery time between tasks in the WB game. This is an important factor because a competitive match is associated with an average duration of 65 min [47].

Considering the analysed context and the FC, the results reported significant differences in most of the variables. On the other hand, the IL variables, such as HR, showed fewer differences between the contexts considering the FC, with the values being higher in competitions. Regarding the neuromuscular EL variables, it was observed the player with FC 2.0 and FC 2.5 performed the highest values in PL/min and impacts/min, respectively. These results are related to those identified in conventional basketball, because the interior players are those with a higher number of impacts, associated with their defensive actions [48].

Differences have been identified in the physical demands between training and competition considering the player's FC, related to the high physical demands associated with competition and the specificity of each position [49]. Therefore, with the aim of improving the agility and speed of the players, it is recommended to develop training exercises focused on explosive strength training programmes [50]. Along the same line, the FC 4.0 and FC 3.0 players present a large range of movements, as well as chair ergonomics and high trunk stability [51], and, because of that, these players should be focused on attack actions, such as throwing, lay-ups, or counterattacks. To improve player performance, coaching staff should consider developing tasks to improve propulsion and Acc techniques [52]. These

techniques improve upper limb function, reducing the incidence of pain as well as the likelihood of injury [24], since the performance of WB players is closely related to muscle strength in the upper limbs, as well as aerobic and anaerobic capacity [53]. Therefore, the coaching staff must take into account those exercises that demand a high physical load from the players depending on the playing position in order that they be administered sufficient rest time to cause specific adaptations that will directly influence the technical-tactical development of the players.

The study has a limitation associated to the small sample size that was used. Working with a national selection can be complex when it comes to scheduling data collection moments. Therefore, for future research, it is recommended to increase the time ranges for data collection. It is also suggested to carry out data collection at different times of the season and compare the performance depending on the moment. However, it should be noted that this study is one of the first to analyse the EL and IL of WB players using IDs. This analysis provides important information to support the understanding of the physical demands of the players.

## 5. Conclusions

Understanding the EL and IL of the players during training and competitions is very important to personalise the training load according to the demands of the competition. It allows development of a progressive and modular training programme of loads to obtain the best performance from the athletes. The coaching staff must develop training sessions that are more intensive and explosive to produce similar physical demands during the training task as in official matches. In this way, the players will develop a great performance during the matches and will increase the probability of winning.

HR is useful for controlling the intensity of workouts. In addition, it is important to design and plan training sessions. However, in WB, players have disabilities and may even have other associated pathologies. Therefore, HR can only be taken as a reference, since it can be affected by various factors such as the athletes' medication, the environment, hydration level, and the duration of exercise. All this can affect the relationship between HR and the training or competition load.

A comprehensive analysis of the physical demands of the athletes will allow the coaching staff to personalise the training loads and identify the intensities at which the players should perform the tasks in training sessions to enhance their performance in competition. Consequently, injury risk of players due to overload will also be reduced.

**Supplementary Materials:** The following supporting information can be downloaded at: https://www.mdpi.com/article/10.3390/app14010269/s1, Tables S1–S5: Descriptive analysis of the EL and IL, and comparison between the context according to the FC.

**Author Contributions:** Conceptualization: V.H.-B., S.J.I. and J.M.G.; methodology: V.H.-B., S.J.I. and J.M.G.; formal analysis: V.H.-B., S.J.I. and J.M.G.; investigation: V.H.-B., S.J.I. and J.M.G.; supervision: V.H.-B., S.J.I. and J.M.G.; data curation: V.H.-B., S.J.I., M.C.E. and J.M.G.; writing—original draft preparation: V.H.-B., S.J.I., M.C.E. and J.M.G.; writing—review and editing: V.H.-B., S.J.I., M.C.E. and J.M.G.; visualization: V.H.-B., S.J.I., M.C.E., and J.M.G.; funding acquisition: V.H.-B., S.J.I., M.C.E. and J.M.G. All authors have read and agreed to the published version of the manuscript.

**Funding:** This research has been partially funded by the project entitled "Scientific-technological support to analyze the training load in basketball teams according to gender, players' level and period of the season" (PID2019-106614GBI00), financed by MCIN/AEI/10.13039/501100011033. The author J.M.G. is a beneficiary of a grant from the Spanish University System Upgrading Programme, Field of Knowledge: Biomedical (Grant Ref.: MS-18). The author M.C.E. is also supported by the Instituto Politécnico de Setúbal and Portuguese Foundation for Science and Technology, I.P., under project number UIDB/04748/2020.

**Institutional Review Board Statement:** The study was conducted in accordance with the Declaration of Helsinki (2013) and approved by the Ethics Committee, University of Extremadura (registration number 233/2019).

**Informed Consent Statement:** Informed consent was obtained from all subjects involved in the study.

**Data Availability Statement:** The data presented in this study are available on request from the corresponding author. The data are not publicly available due to privacy restrictions.

**Acknowledgments:** This study was developed within the Optimization of Training and Sports Performance Research Group of the Faculty of Sports Science of the University of Extremadura. All authors have contributed to the study, and it is certified that it is not under consideration for publication in another journal. The authors would like to acknowledge the participants who allowed us to conduct this study.

**Conflicts of Interest:** The authors declare no conflict of interest.

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
