# Peer review of "Analysis of the External and Internal Load in Wheelchair Basketball Considering the Game Situation"

_applsci, doi:10.3390/app14010269_

Round 1

Reviewer 1 Report

Comments and Suggestions for Authors

Abstract

- Line 26 - Higher than? External and internal load?

- In the method, I suggest including data values and statistical analysis if you have space.

- Lines 28 to 31 - I understand that this part is the practical application of the study. However, what is the conclusion based on the main results?

Introduction

- Line 35 - Does ID quantify internal load or just external load?

- Line 39 - I suggest focusing specifically on Paralympic sport, and I leave the article by Stieler et al. as a suggestion. About external load assessment methods in Paralympic sports (https://doi.org/10.1123/jsr.2022-0110).

Methods

- Why are participants and samples separated? Wouldn't it be better to be in the same item?

- Were there defined inclusion and exclusion criteria?

- Line 102 - Was the complete game replicated, simulating official matches? How it is written suggests that he had tasks similar to those of the official match.

- Table 1 - The table contains only 9 players. Is this correct? Weren't there 12?

- Line 136 - Check formatting.

- Line 142 - Is the Shapiro-Wilk test not recommended for samples smaller than 30 subjects?

- Line 146 - To make this comparison, did the authors group the data from the two matches?

- Lines 148 and 149 - What do you mean by type of task?

Results

- Table 2 - Insert what X and SD are in the legend. Also, is it OK to report through X?

- Line 170 – (Figure 1a) I suggest including whether it was in the external and/or internal load parameters, as it would be too extensive to include each variable that made a difference.

- Line 176 – "higher FC" How much higher? Until now, you have used the classification numbering; I suggest including it here, too.

- Figure 1 - Different font sizes and values within the figure (parts A and B).

Discussion

- Lines 184 and 185 - Wouldn't it be possible to analyze the influence on the external and internal load? I think that to see the influence on sporting performance, other variables could be investigated, such as the number of successful shots, for example.

- Line 218 - What results? If you are going to discuss beats per minute, bring this data from the present study.

- Lines 219 and 220 - If the study averaged 131 and 152 beats, wouldn't it be similar to the 140 beats?

Conclusion

- In my view, it seems that the conclusion was similar to the practical applications of the study. My suggestion is to include the practical applications of the study after the limitations and finally, in the conclusion, respond to the proposed objective with the main results.

Reviewer 2 Report

Comments and Suggestions for Authors

The article ”Analysis of the External and Internal Load in Wheelchair Basketball Considering the Game Situation” has a fascinating topic, and the authors have tried to demonstrate that analyzing the external and internal load of wheelchair basketball players during training and competitive matches, taking into account the functional classification of players.

The introduction is well done, but the authors should in a few sentences explain the characteristics of the athletes with spinal cord injury, amputees, spina bifida, and joint and muscle limitations

The authors did not interpret the results clearly and the graphs obtained in Fig. 1 (Comparison of the heart rate regarding the context analyzed) have no logical correlation with the paper itself.
The section interpreting the results is limited to 10 lines 168-178, which is contrary to the rules of presentation of a scientific article that is statistically evaluated.
The conclusions do not reflect the discussions nor do the results cover them.
Unfortunately, the article cannot be published in this form.

Round 2

Reviewer 1 Report

Comments and Suggestions for Authors

I appreciate the authors' responses and am pleased with the review.

Reviewer 2 Report

Comments and Suggestions for Authors

After the changes made, the article can be published